# Building Large Machine Learning Models from Small Distributed Models: A Layer Matching Approach

## Abstract

Cross-device federated learning (FL) enables massive amount of clients to collaborate to train a machine learning model with local data. However, the computational resource of the client devices restricts FL from utilizing large modern machine learning models that requires sufficient computation. In this paper, we propose a federated layer matching algorithm that enables the server to build a deep server machine learning model from relatively shallow client models. The federated layer matching (FLM) algorithm dynamically averages similar layers in the client models to the server model, and inserts dissimilar layers as new layers to the server model. With the proposed algorithm, the clients are able to train small models based on device capacity, while the server can still obtain a larger and more powerful server model from the clients with decentralized data. Our numerical experiments show that the proposed FLM algorithm is able to build a server model $40\%$ larger than the client models, and such a model performs much better than the model obtained by the classical FedAvg, when using the same amount of communication resource.

## 1 Introduction

Machine learning has been widely used in various applications, such as computer vision (CV) and natural language processing (NLP). To process large amounts of complex data (image, text, time-series, graph), machine learning (ML) models, especially deep learning models, have become extremely large in their widths and depths. For example, GPT-3[1] has 175 billion model parameters with 96 layers; RegNetY [2] has 145 million model parameters and 62 layers; ViT [3] has 632 million model parameters with 32 layers.

Even though large models have become state-of-the-art for many application domains, there are several critical drawbacks to building these models. First of all, training such huge models requires tremendous amounts of data as well as computation resources. Second, in many applications (such as those in federated learning), there is a massive amount of data available but distributed among mobile devices such as laptops, tablets, and mobile phones, which have limited bandwidth, memory, and computation resources. As a result, it becomes impossible to deploy and train huge ML models on resource-constrained devices.

In this work, we propose a novel approach capable of pooling distributed and heterogeneous resources (both data and computation) to train a large ML model. This is made possible by making the key observation that modern ML models often have certain *stacked* structures. That is, by repetitively stacking certain components on top of each other, one can build larger and larger models. For example, ResNet-50 repeats the third Bottleneck layer 6 times while ResNet-101 repeats the same layer 23 times [4]; Bert-small stacks 4 identical attention layers while Bert-large stacks 24 layers [5]. With this observation, we ask and hope to address the following research question:

**(Q)** Is it possible to progressively and efficiently aggregate small client-side models to construct deep and powerful server-side models, without degrading model performance?

**Our contributions.** In this project, we address the above question **(Q)** by considering the situation that many clients are connected to a server, and the clients possess private data and some computation power. The goal is to collectively build a large and deep model by utilizing as much local data and computation power as possible while minimizing the communication overhead in the system. Towards this end, we propose a *federated layer matching* (FLM) algorithm, which is, to our knowledge, the first method capable of dynamically constructing a large and deep neural network (located at the server) from shallow ones (located at the clients), by using data distributed over the clients. The proposed algorithm enjoys many desirable features, such as low communication cost and strong theoretical guarantees. More specifically, the main contributions of this work are summarized below.

- **From shallow to deep.** To our knowledge, this is the first work that enables distributed and systematic construction of a *deep* neural network from *shallow* ones, while the closest existing works, such as FedMA [6], can only construct *wider* networks with *fixed* depth.

- **Flexibility.** The proposed method has been carefully designed so that it is *flexible*, in the sense that it can be used to build a wide range of different neural network structures encountered in CV and NLP applications. This is made possible by the novel design of certain *layer matching* mechanism, to be introduced in Section 2.2.

- **Communication efficiency.** In contrast to the existing (horizontal) FL algorithms such as FedAvg, which require frequent exchange of the *entire* model between the clients and the server, the proposed method is communication efficient because it can better utilize the heterogeneous client models by the proposed *layer matching* technique, rather than simply averaging them.

- **Superior empirical performance.** Finally, we conduct empirical experiments on ResNet with Cifar-10 dataset, showing that the proposed algorithm has better performance (measured by prediction accuracy, communication/sample efficiency, etc.) compared with state-of-the-art (SOTA) algorithms. In certain cases, its performance can even approach that of centralized training algorithms.

## 2 Preliminary and Problem Setup

### 2.1 Problem Setup

Consider the distributed problem with loss function $f(\cdot)$ and $K$ clients. Suppose each client has dataset $\mathcal{D}_k$. We aim to solve the following problem:

$$\min_{\boldsymbol{\Theta}} \sum_{k=1}^{K} f(\boldsymbol{\Theta}; \mathcal{D}_k), \tag{1}$$

where $\boldsymbol{\Theta}$ denotes model parameters. To model the *stacked* structure of the ML models, let us assume that the client $k$'s model parameters, denoted by $\boldsymbol{\Theta}_k$, consists of $M$ blocks $\boldsymbol{\Theta}_k := [\boldsymbol{\Theta}_{k,1}; \ldots; \boldsymbol{\Theta}_{k,M}$, where $\boldsymbol{\Theta}_{k,m}$ represents the parameters of the $m$th block. Each block $m$ is stacked with $L_{k,m}$ layers of the *same* structure. Thus, the parameter of the $m$th block can be further written as $\boldsymbol{\Theta}_{k,m} = [\boldsymbol{\Theta}_{k,m}[1]; \cdots ; \boldsymbol{\Theta}_{k,m}[L_{k,m}]]$, where $\{\Theta_{k,m}[l]\}_{l=1}^{L_{k,m}}$ have the same dimension, and the model of client $k$ has $L_k = \sum_{m=1}^{M} L_{k,m}$ layers in total. In addition, let us use $0$ as the index of the server, e.g., the server model is denoted as $\boldsymbol{\Theta}_0$.

Federated Learning (FL) studies the setting of solving (1) in a parameter-server/client system. FedAvg [7] is a well-studied solution of FL that adopts the computation-then-aggregation strategy. In the algorithm, the clients $k = 1, \ldots, K$ locally perform a few steps of model updates by optimizing:

$$\min_{\boldsymbol{\Theta}_k} f(\boldsymbol{\Theta}_k; \mathcal{D}_k). \tag{2}$$

Then, the server aggregates the updated local models and averages them before sending the updated global model back to the clients, that is, it performs:

$$\boldsymbol{\Theta}_0 = \frac{1}{K} \sum_{k=1}^{K} \boldsymbol{\Theta}_k.$$

However, due to the limited communication and computation resource on the mobile devices, typical FL algorithms can only be used to train small models such as MobileNet or MLPs. These models have few parameters than the SOTA models and have sub-optimal performance.

A more recent approach is to ensemble the client models to construct a larger server model [8, 6, 9], which aims to train small models on the clients and ensemble the client models into a large server model. In this case, the clients still solve (2) locally with multiple updates. Then, the server aggregates the client models $\{\mathbf{\Theta}_k\}_{k=1}^K$ and builds the server model $\mathbf{\Theta}_0$ by a linear or non-linear transformation, denoted by

$$\mathbf{\Theta}_0 = \text{Agg}(\mathbf{\Theta}_1, \ldots, \mathbf{\Theta}_K).$$

In these approaches, the server model $\mathbf{\Theta}_0$ has more parameters than the client models $\mathbf{\Theta}_k$.

However, this line of work still has scalability, client resource requirement, and computation efficiency limitations. In our work, we focus on a new model ensemble approach: the *layer matching* technique. The key idea of our approach is to train shallow models in the clients based on the client resource capacity, and the server properly stacks the layers of the client model into a deeper server model. Compared with FedAvg, our approach can train a deeper server model with better performance and the ability to fully utilize the massive data and limited computation resources on a large number of clients.

## 2.2 Related Work

**Federated Learning:** The FL problems typically consider the setting that the clients are directly connected to a parameter-server and that the communication at the server is the bottleneck of the system. The FL algorithms, such as the well-known FedAvg [7], perform multiple local updates before one communication step. However, when the data is *heterogeneous* among the agents, it is difficult for these algorithms to achieve convergence [10, 11]. Recent algorithms such as the FedProx [12], SCAFFOLD [13] and FedPD [14] have developed new techniques to improve upon FedAvg. These algorithms require the server and client models to have the same size to perform model averaging. This requirement restricts the algorithms from training large ML models when (a part of) the clients have limited computation capacity.

**Model Ensembling:** Model ensemble method have been widely used in FL [8, 15, 16]. They are used to construct server models out of collected client models. In [17, 18], the authors propose a straightforward method that directly concatenates local models into a wide model. The server model obtained by the above-mentioned *stacking* method scales *linearly* with the client number, and unfortunately, it omits the interconnection among the client models. These properties can result in a waste of memory and computation resource and have less scalability. A perhaps smarter way of performing the model ensemble is the neural matching method [6]. This method tries to identify the similarity in hidden elements (e.g., hidden states in LSTM, convolution channels in CNN) and match them in different client models and construct a wider server model. Note that, this method only matches the parameters in the same layer to extend the width of the aggregated model and preserves the depth of the client models. However, many studies have theoretically and empirically suggested that the depth of the neural network is critical in strengthening the network representation ability and increasing network performance [19, 20].

# 3 Federated Layer Matching

## 3.1 Algorithm Description

In this section, we discuss the proposed layer matching algorithm. The key idea is that, the server matches the layers in different client models based on the *similarity* of the layer's parameters instead of the layer's position. In specific, for each block $m$ (i.e., a residual block in ResNet), the server first aggregates a total of $\sum_{k=1}^K L_{k,m}$ layers of the same structure. It then clusters these layers by the similarity between their parameters $\mathbf{\Theta}_{k,m}[l]$, and such clustering operation will result in $L_{0,m}$ clustered layers, which can be deeper than the client models, i.e., $L_{0,m} \geq \max\{L_{k,m}\}_{k=1}^K$. By stacking the clustered layers, the server can construct a deeper server model.

Before going into the detailed explanation of the algorithm, let us first define some notations for convenience. First, let us define the layer matching pattern for client $k$ as a matching matrix $\Pi_k \in$

---

**Algorithm 1** Federated Layer Matching Algorithm

---

1: **Inputs:** Data $\mathcal{D}_k$, matching frequency $R$, maximal iteration $T$, initial model $\Theta_0^0$.
2: **Initialization:** The server broadcasts the initial server model $\Theta_0^0$ to all clients. $\Pi_k^0 = I$
3: **for** $t = 1, \ldots, T$ **do**
4:   **for** Client $k = 1, \ldots, K$ in parallel **do**
5:     **Step 1**: Optimize $\Theta_k^t = \arg\min_{\Theta_k} \mathcal{L}_k(\Theta_k; \Theta_0^{t-1})$;
6:     Send updated model $\Theta_k^t$ to server.
7:   **end for**
8:   **Server:**
9:   **if** $t \mod R = 0$ **then**
10:     **Step 2**: Solves the matching problem (4) and obtains the layer matching pattern $\Pi_k^t$
11:   **else**
12:     Uses the matching pattern at the previous round, namely to let $\Pi_k^t = \Pi_k^{t-1}$
13:   **end if**
14:   **Step 3**: Construct the server model: $\Theta_0^t = \frac{\sum_{k=1}^K \Pi_k^t \Theta_k^t}{\sum_{k=1}^K \Pi_k^t \mathbb{1}_{L_k}}$.
15:     **Step 4**: Send the updated server model and matching patterns $\Pi_k^t{}^\top \Theta_0^t$ to the clients
16: **end for**
17: Return: Parameters $\Theta_0^T$.

---

130  $\{0,1\}^{L_0 \times L_k}$, where $\Pi_k[l_0, l_k] = 1$ if layer $l_k$ of the client model matches layer $l_0$ of the server model
131  and $\Pi_k[l_0, l_k] = 0$ if layer $l_k$ of the client model does not match layer $l_0$ of the server model. As
132  the layers can only be matched to the layer in the same block, the matching matrix must be a block
133  diagonal matrix with $M$ blocks $\Pi_k = \text{diag}\{\Pi_{k,1}, \ldots, \Pi_{k,M}\}$, where $\Pi_{k,m} \in \{0,1\}^{L_{0,m} \times L_{k,m}}$.

134  Further, we define the local loss function as

$$\mathcal{L}_k(\Theta_k; \Theta_0) \triangleq f(\Theta_k; \mathcal{D}_k) + r(\Theta_k, \Theta_0),$$

135  where $r$ is a regularizer defined as:

$$r(\Theta_k, \Theta_0) \triangleq \left\| \Theta_k - \Pi_k{}^\top \Theta_0 \right\|_2^2.$$

136  Finally, let $\mathbb{1}_L := [1; 1; \cdots ; 1] \in \mathbb{R}^L$ denote the all-one vector of size $L$.

137  The algorithm at each round of training consists of the following four major steps. Algorithm 1
138  provides a detailed description of the algorithm.

139  • Step 1: Each client $k \in [K]$ updates local model $\Theta_k$ by optimizing the local loss function $\mathcal{L}_k$, and
140    sends the local model to the server;

141  • Step 2: The server cluster the layers in the aggregated client models based on the similarity be-
142    tween the layer parameters. By optimizing the layer matching problem (4), the server obtains the
143    layer matching patterns $\Pi_k$ for all clients;

144  • Step 3: The server constructs the server model by stacking the clustered layers and reconstruct the
145    client models with the server model and the layer matching patterns;

146  • Step 4: The server sends the reconstructed client models to each client and starts the next round
147    of training.

148  Next, let us provide a detailed explanation to the key step 2 in the proposed algorithm. More detailed
149  description of the algorithm are given in Appendix A.

150  **Step 2:** The server-side layer matching is the key step for the proposed FLM algorithm. At the iter-
151  ation to perform layer matching $t \mod R = 0$, for each block $m \in [M]$ in parallel, we sequentially
152  match each client layers to the server layers for $P$ iterations. The matching procedure is illustrated
153  in Algorithm 2, which consists four stages, and the major stages 2 and 3 are discussed below.

154  **Stage 2-2**: In this stage, we compute the cost for each client layer $l_k = 1, \ldots, L_{k,m}$ in the block to
155  a layer in the server. Assume that we set the maximum number of layers in block $m$ in the server

---

**Algorithm 2** Layer Matching Step 2

---
1: **Inputs:** Client models $\{\mathbf{\Theta}_k^t\}_{k=1}^K$, old server model $\mathbf{\Theta}_0^{t-1}$, and matching patterns $\{\Pi_k^{t-1}\}_{k=1}^K$.
2: **Initialize:** $\Pi_k^{t,0} = \Pi_k^{t-1}, \forall\, k \in \{1, \ldots, K\}$
3: **for** Block $m = 1, \ldots, M$ in parallel **do**
4:     **for** $p = 0, \ldots, P-1$ **do**
5:       Randomly select client $k$ in $\{1, \ldots, K\}$ at random;
6:       **Stage 2-1**: Construct server model $\mathbf{\Theta}_{0,m}^{t,p} = \frac{\sum_{k' \neq k} \Pi_{k',m}^{t,p} \mathbf{\Theta}_{k',m}^t}{\sum_{k' \neq k} \Pi_{k',m}^{t,p} \mathbb{1}_{L_{k',m}}}$
7:       **Stage 2-2**: Compute cost matrix $\mathbf{C}_{k,m}^t$ matching $\mathbf{\Theta}_{k,m}^t$ to $\mathbf{\Theta}_{0,m}^{t,p}$ with (3);
8:       **Stage 2-3**: Solve matching pattern $\Pi_{k,m}^{t,p+1}$ of block $m$ of client $k$ by optimizing (4);
9:       **Stage 2-4**: Update all other matching patterns $\{\Pi_{k',m}\}_{k' \neq k}$.
10:     **end for**
11: **end for**
12: Return: Updated matching patterns $\{\Pi_k^t = \Pi_k^{t,P}\}_{k=1}^K$.

---

model as $\bar{L}_{0,m}$. Then the costs for matching all layers in block $m$ for client $M$ form a cost matrix $\mathbf{C}_{k,m}^{t,p} \in \mathbb{R}^{\bar{L}_{0,m} \times L_{k,m}}$, where its entries are defined as:

$$\mathbf{C}_{k,m}^{t,p}[l_0, l_k] = c_1(\mathbf{\Theta}_{k,m}^t[l_k], \mathbf{\Theta}_{0,m}^{t,p}[l_0]) + c_2(\mathbf{\Theta}_{k,m}^t[l_k], l_0, \bar{L}_{0,m}), \tag{3}$$

where $c_1(\cdot)$ is the *fidelity score* measures the similarity between the client layer and the server layer, and $c_2(\cdot)$ is the *model complexity penalty* for increasing the number of the server layer. A reasonable choice for $c_1(\cdot)$ is a distance metric (e.g., Euclidean distance or cosine angle) between the server and client layers, and the choice for $c_2(\cdot)$ can be the model selection metric (e.g., $K \log(L_{l_0})$ used in BIC) on the model size.

**Stage 2-3** With the cost matrix $C_{k,m}^{t,p}$ that describes the cost to match each client layer to the server layers, we can optimize the corresponding linear assignment problem:

$$\min_{\Pi_{k,m}^{t,p+1}} \text{Trace}\left( (\Pi_{k,m}^{t,p+1})^\top \mathbf{C}_{k,m}^{t,p} \right), \quad \text{s.t. } \Pi_{k,m}^{t,p+1} \in \{0,1\}^{\bar{L}_{0,m} \times L_{k,m}}, (\Pi_{k,m}^{t,p+1})^\top \mathbb{1}_{\bar{L}_{0,m}} = \mathbb{1}_{L_{k,m}}, \tag{4}$$

and obtain the updated matching pattern $\Pi_{m,k}^{t,p+1}$. If any of the rows $L_{0,m}+1, \ldots, \bar{L}_{0,m}$ in the matching pattern are not all-zero, then some of the client layers are added to the server model as new layers, and the server model becomes deeper. Otherwise, the server model keeps the same depth $\bar{L}_{0,m}$ or even gets shallower.

By repeating Stage 2-1 to 2-4 for $P$ iterations ($P \geq K$), all the client matching patterns $\{\Pi_k^t\}_{k=1}^K$ will be updated. This concludes the entire procedure of Step 2 in Algorithm 1.

**Step 3**: In this step, the server constructs the server model with all the updated client matching patterns $\{\Pi_k^t\}_{k=1}^K$ and the client models $\{\mathbf{\Theta}_k^t\}_{k=1}^K$, by averaging the client layers that match to the same server layer, i.e.,

$$\mathbf{\Theta}_{0,m}^t[l_0] = \frac{1}{\left| \{l_k | \Pi_{k,m}^t[l_0, l_k] = 1\} \right|} \sum_{l_k \in \{l_k | \Pi_{k,m}^t[l_0, l_k]=1\}} \mathbf{\Theta}_{k,m}^t[l_k].$$

Now the server model has been constructed, the server then reconstructs the client model with $\mathbf{\Theta}_k^t = \Pi_k^t{}^\top \mathbf{\Theta}_0^t$. Such an updated client model will then be used for the local training at the next iteration $t+1$.

## 3.2 Discussion

Before we close this section, let us provide some discussions about the algorithm.

First, let us discuss the computation/communication resources used at the clients and the servers. In each round of training, each client trains a shallow network with parameter $\mathbf{\Theta}_k$, sends the shallow

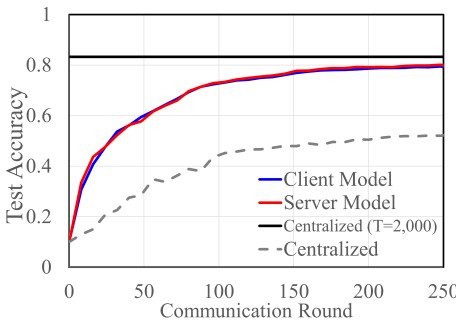 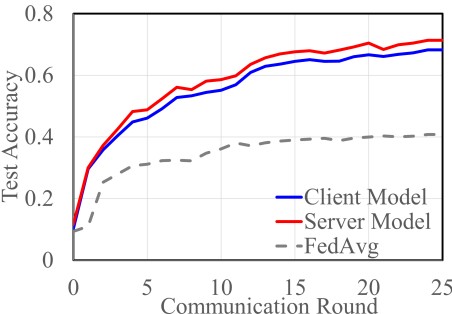

(a) Testing accuracies of the client models, the server model, centralized trained model with $T = 2,500$ iterations, and the centralized trained model with the same number of updates.

(b) Testing accuracies of the client models, the server model trained with FLM, and the model trained with FedAvg with the same number of communication rounds.

Figure 1: Test accuracy of a) IID distributed data, b) Non-IID distributed data on Cifar-10 dataset.

network $\boldsymbol{\Theta}_k^t$ and receives the model $\Pi_k^{t\top}\boldsymbol{\Theta}_0^t$. The resource cost on the client is the same as Fed-Prox [12]. The server needs to solve the layer matching problem described in **Step 3**, which has computation complexity of $\mathcal{O}(K \times \sum_{m=1}^M N_m^3 \times P)$, and scales linearly with the number of clients.

Second, let us remark that although the algorithm appears to be rather complicated, in fact its mathematical interpretation is relatively simple. Instead of directly solving (1), we solve the following *regularized* problem:

$$\min_{\boldsymbol{\Theta}_0, \{\boldsymbol{\Theta}_k, \Pi_k\}_{k=1}^K} \frac{1}{K} \sum_{k=1}^K \left( f(\boldsymbol{\Theta}_k; \mathcal{D}_k) + \left\| \boldsymbol{\Theta}_k - \Pi_k^{\top} \boldsymbol{\Theta}_0 \right\|^2 \right). \tag{5}$$

Note that not only the client/server parameters $\boldsymbol{\Theta}_k$'s are $\boldsymbol{\Theta}_0$ are optimized, the matching pattern $\Pi_k$'s are also getting optimized. The client and the server alternatingly optimize $\{\boldsymbol{\Theta}_k\}$ and $\boldsymbol{\Theta}_0, \{\Pi_k\}$ respectively by using block coordinate descent (BCD)-type algorithm. In the server-side optimization, the shape of $\Pi_k$ and $\boldsymbol{\Theta}_0$ are dynamically changing based on the heterogeneity of the layers in the client models. Following the line of the proof in [21], we can show that the FLM algorithm generates $\boldsymbol{\Theta}_0, \{\boldsymbol{\Theta}_k, \Pi_k\}_{k=1}^K$ that converge to the first-order stationary point of (5) with rate $\mathcal{O}(1/T)$.

# 4 Numerical Experiments

We run the experiment based on ResNet model designed for Cifar-10 dataset [22]. In the experiment, we use ResNet$[a, b, c]$ to denote the number of residual layers in the three residual blocks in the ResNet model. The dataset is distributed among the clients following one of the two cases: 1) IID case: the samples are uniformly distributed to the clients; 2) Non-IID case: $50\%$ of the samples on each client belong to two classes while the other $50\%$ of the samples are uniformly picked from the rest of the classes. The detailed descriptions of the experiments are included in Appendix B.

## 4.1 Numerical Results

In the IID case, we compare the performance of the models obtained with FLM algorithm on $K = 8$ clients with centralized training. The result is shown in Figure 1a. In this setting, the client models use ResNet$[5, 5, 5]$ and the matched server model has layer number $[5, 5, 6]$. We can see that the performance of the trained client models and that of the matched server model are comparable to the centralized model, but the required number of local update is much less than centralized training.

In the Non-IID case, we compare the performance of the models obtained with FLM algorithm and FedAvg algorithm with $K = 5$ clients. The result is shown in Figure 1b. The performance of the server model, as well as that of the client models trained with FLM under limited number of communication round are much better than the model trained with FedAvg. The matched server model has layer $[6, 7, 8]$, and the client models are ResNet$[5, 5, 5]$. In the experiment, FLM can better utilize the heterogeneous layers from different clients to build $40\%$ deeper server model to deal with data heterogeneity, while FedAvg only averages the client models.

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

# A  Algorithm Details

In this section, we provide the detailed explanation of step 2 and 3 in Algorithm 1.

For clarity of the notations, we put a list of notations with their meanings in Table 1.

| Notation | Notation meaning |
|---|---|
| $(\cdot)_0$ | Server index |
| $(\cdot)_k$ | Client index |
| $K$ | Total client number |
| $(\cdot)^t$ | Iteration index |
| $T$ | Total training iteration |
| $(\cdot)^p$ | Matching index |
| $P$ | Total layer matching iteration |
| $R$ | Layer matching frequency |
| $(\cdot)_m$ | Block index |
| $M$ | Total block number |
| $(\cdot)[l]$ | Layer index |
| $L, \bar{L}$ | Total layer number, maximum layer number |
| $\Theta$ | Model parameter |
| $\Pi$ | Matching pattern |

Table 1: Notation Table

**Step 2:** The server-side layer matching is the key step for the proposed FLM algorithm. At the iteration to perform layer matching $t \mod R = 0$, for each block $m \in [M]$ in parallel, we sequentially match each client layers to the server layers for $P$ iterations. The matching procedure is illustrated in Algorithm 2, which consist four major stages discussed below.

**Stage 2-1**: After randomly choosing a client $k$ to perform layer matching, the first stage is to construct the server model $\Theta_{0,m}^{t,p}$ with the layers of the other clients and their matching patterns, i.e.,

$$\Theta_{0,m}^{t,p} = \frac{\sum_{k' \neq k} \Pi_{k',m}^{t,p} \Theta_{k',m}^t}{\sum_{k' \neq k} \Pi_{k',m}^{t,p} \mathbb{1}_{L_{k',m}}},$$

which removes the impact of the parameters of client $k$. Therefore, the layer similarities between the client and server models computed in the following matching stages is not affected by the previous matching pattern $\Pi_{k,m}^{t-1}$ of client $m$.

**Stage 2-2**: In this stage, we compute the cost for each client layer $l_k = 1, \ldots, L_{k,m}$ in the block to a layer in the server. Assume that we set the maximum number of layers in block $m$ in the server model as $\bar{L}_{0,m}$. Then the costs for matching all layers in block $m$ for client $M$ form a cost matrix $\mathbf{C}_{k,m}^{t,p} \in \mathbb{R}^{\bar{L}_{0,m} \times L_{k,m}}$, where its entries are defined as:

$$\mathbf{C}_{k,m}^{t,p}[l_0, l_k] = c_1(\Theta_{k,m}^t[l_k], \Theta_{0,m}^{t,p}[l_0]) + c_2(\Theta_{k,m}^t[l_k], l_0, \bar{L}_{0,m}), \tag{6}$$

where $c_1(\cdot)$ is the fidelity score to match a client layer to an existing server layer and $c_2(\cdot)$ is the penalty to increase the number of the server layer. A reasonable choice for $c_1(\cdot)$ is a distance metric (e.g., Euclidean distance or cosine angle) between the server and client layers, and the choice for $c_2(\cdot)$ can be model selection metric (e.g., BIC) on the model size.

**Stage 2-3** With the cost matrix $C_{k,m}^{t,p}$ that describe the cost to match each client layer to the server layers, we can optimize the corresponding linear assignment problem:

$$\min_{\Pi_{k,m}^{t,p+1}} \text{Trace}\left( (\Pi_{k,m}^{t,p+1})^\top \mathbf{C}_{k,m}^{t,p} \right), \quad \text{s.t. } \Pi_{k,m}^{t,p+1} \in \{0,1\}^{\bar{L}_{0,m} \times L_{k,m}}, (\Pi_{k,m}^{t,p+1})^\top \mathbb{1}_{\bar{L}_{0,m}} = \mathbb{1}_{L_{k,m}}, \tag{7}$$

and obtain the updated matching pattern $\Pi_{m,k}^{t,p+1}$. If any of the rows $L_{0,m} + 1, \ldots, \bar{L}_{0,m}$ in the matching pattern are not all-zero, then some of the client layers are added to the server model as new layers, and the server model becomes deeper. Otherwise, the server model keeps the same depth $\bar{L}_{0,m}$ or even gets shallower.

**Stage 2-4**: After updating the matching pattern from $\Pi_{m,k}^{t,p}$ to $\Pi_{k,m}^{t,p+1}$, the server layer order and the total layer number may change. So in this stage, we need to update the matching patterns of the other clients $\{\Pi_{k',m}^{t,p+1}\}_{k'\neq k}$ accordingly. First, we compute the "average layer index" for each server layer $l_0^{t,p}$:

$$\bar{l}_0 = \frac{1}{\left|\{l_k|\Pi_{k,m}^t[l_0,l_k]=1\}\right|}\sum_{l_k\in\{l_k|\Pi_{k,m}^t[l_0,l_k]=1\}} l_k.$$

Then, we reorder the server layers based on the order of the average layer index $\bar{l}_0$ and obtain the new server layer indices $l_0^{t,p+1}$. Finally, the layer matching patterns for the clients are updated with $\Pi_{k,m}^{t,p+1}[l_0^{t,p+1},l_k] = \Pi_{k,m}^{t,p}[l_0^{t,p},l_k]$.

By repeating Stage 2-1 to 2-4 for $P$ iterations ($P \geq K$), all the client matching patterns $\{\Pi_k^t\}_{k=1}^K$ will be updated and that finishes the whole procedure of Step 2 in Algorithm 1.

**Step 3**: In this step, the server construct the server model with all the updated client matching patterns $\{\Pi_k^t\}_{k=1}^K$ and the client models $\{\boldsymbol{\Theta}_k^t\}_{k=1}^K$ as

$$\boldsymbol{\Theta}_0^t = \frac{\sum_{k=1}^K \Pi_k^t \boldsymbol{\Theta}_k^t}{\sum_{k=1}^K \Pi_k^t \mathbb{1}_{L_k}},$$

which averages the client layers that match to the same server layer, i.e.,

$$\boldsymbol{\Theta}_{0,m}^t[l_0] = \frac{1}{\left|\{l_k|\Pi_{k,m}^t[l_0,l_k]=1\}\right|}\sum_{l_k\in\{l_k|\Pi_{k,m}^t[l_0,l_k]=1\}} \boldsymbol{\Theta}_{k,m}^t[l_k].$$

With the server model, the server then reconstruct the client models with $\boldsymbol{\Theta}_k^t = {\Pi_k^t}^\top \boldsymbol{\Theta}_0^t$ that serves as the initial client model and the regularizer for the local training at the next iteration $t+1$.

# B  Experiment Details

**Model:** The structure of the model is summarized in Table 2. Each residual block is constructed with two Conv2d layers, each followed by one GroupNorm layer [23] and a ReLU activation layer, and finally add an identical mapping to the end. In the experiment, we use ResNet$[a,b,c]$ to denote the ResNet model using certain number of residual blocks of block 2, 3 and 4 in the table. During model aggregation phase, we perform layer matching to the parameters in block 2, 3B and 4B separately, and average the parameters in the other layers.

| Block # | Shape | Parameter # |
|---|---|---|
| 1 | Conv2d($3 \times 3 \times 16$) | 448 |
| | BatchNorm2d(16) | – |
| 2 | ResidualLayer($16, 16$) | 4,608 |
| 3A | ResidualLayer($16, 32$) | 13,824 |
| 3B | ResidualLayer($32, 32$) | 18,432 |
| 4A | ResidualLayer($32, 64$) | 55,296 |
| 4B | ResidualLayer($64, 64$) | 73,728 |
| 5 | AvgPool() | – |
| | Linear($64 \times 10$) | 650 |

Table 2: ResNet model structure for Cifar-10 dataset.

**Dataset:** In the experiment, we use Cifar-10 dataset. There are two data distribution settings: 1) IID setting: in this setting the samples are uniformly distributed to clients; 2) Non-IID setting: in this setting, $50\%$ of the samples on each client belongs to two classes while the other $50\%$ of the samples are uniformly picked from the rest of the classes.

