# OpenReview forum: "Building Large Machine Learning Models from Small Distributed Models: A Layer Matching Approach"
_NeurIPS.cc/2022/Workshop/Federated_Learning — FL-NeurIPS 2022 Poster_

### Official Review · Reviewer_XSuM · 2022-10-14

The paper proposes a new federated learning framework to enable the server model to be directly constructed by the client model. In other words, the client models could maintain shallow while the sever model is larger and powerful. Specifically, it proposes the federated layer matching algorithm to penalize those layers in client model similar with server models and, at the same time, regularize the increase of the server model layers. The experiments results show the proposed method could achieve a good result.

Pros:
1. The proposed framework is very interesting and novel. It could help to address key bottlenecks in the current FedML framework.
2. The paper is well-written and easy to follow.
3. The experiment shows the proposed method could achieve a good result.

Cons:
1. It is not obvious to me how the proposed method could used in other architecture rather than ResNet. For example. the kernel size, pooling could be totally different for some convolutional neural networks. Also the hyper meter is different in the ViT.
2. The performance of the baseline is a little too bad. I think it should be easy to get over 90% accuracy using FedAvg in the CIFAR10 dataset.

---

### Official Review · Reviewer_2k1r · 2022-10-17
**A promising idea with many missing details**

This work explores how to utilize federated learning techniques to learn a large model across clients, even when clients can only train smaller models (eg. due to resource limitations). In particular, it proposes a layer matching technique in which similar client layers are averaged together, and non-matching layers effectively become new layers in the server model. Finally, the work gives some simulations of the method on CIFAR-10.

As the title of this review suggests, I think that this idea is promising. Fundamentally, resource limitations in FL are to date large obstacles in training models. Their size impacts everything from broadcast, to upload, to local client memory usage. Moreover, so-called "scaling laws" of neural networks seem to suggest that larger models perform better in a myriad of ways. Thus, the idea presented in this work has some promise, especially as the authors attempt to remove restrictions of prior works that essentially did no grouping among client layers when building the server layer. The idea of matching client updates according to similarity has recently seen a lot of interest, and it seems on its face valuable to incorporate this into this layer matching approach.

That being said, there are a number of missing details (especially in the empirical evaluation) that lead me to believe that the work is not fully descriptive and reproducible. Moreover, this lack of details prevents the reader from understanding what core innovations are actually driving the improvement of the method (if there is any, see discussion below).

### Missing Detail #1: Which federated learning setting is being studied?

The first missing details (though this is probably a misnomer in this case) is a lack of examination of what federated setting is even being considered here. Specifically, the authors seem to be considering a case where clients are restricted to smaller models, and yet seem to assume that the same set of clients are queried in each communication round, and are always available. Unfortunately, the combination of these two characteristics is relatively unrealistic. While a set of resource constraints tends to occur in cross-device settings (see [Kairouz et al., 2019]), this is paired with relatively unreliable participation and a large number of total clients. The end result is that in cross-device settings, devices tend to participate only a few times throughout training (if at all). By contrast, in cross-silo settings, where clients have more reliable participation (and there are only a small number of them), resource constraints are much more relaxed.

This is all to say that to really motivate the algorithm, I think it would need to contend with things like partial/limited participation of clients. In particular, what should clients who are participating for the first time do? What matching matrix pattern should they receive? What if we cannot identify the clients at all? These are all questions that should be asked in cross-device settings. While I understand that not every paper can answer all questions simultaneously, the experiments have a small enough number of clients that I can't tell whether the method has a chance of being useful in cross-device settings.

### Missing Detail #2: How are the experiments being performed?

The other crucial missing details from the work are how the experiments are being performed. While Section 4 gives a brief sketch of the experimental setting, many aspects are unclear. First and foremost, it is not clear what accuracies are being plotted in Figure 1. While the test accuracy of the server model is relatively standard, it is unclear what the "client model" (note the singular) refers to. Are the authors plotting the average accuracy of all client models? If so, do the results not seem to suggest that each client could essentially optimize just its own local model without needing to do any communication?

Other aspects of the experimental results seem confusing at best, potentially misleading at worst. For example, comparing Figures 1a and 1b, we see that in the Non-IID case, the server model can obtain roughly a 60% accuracy in 10 communication rounds. In the IID case it takes nearly 50 communication rounds. This defies the notion that heterogeneous data hurts optimization. While the obvious counterpoint is the server model sizes (the authors use Resnet[5, 5, 6] for IID data and Resnet[6, 7, 8] for non-IID) this begs the question of why the authors used different model sizes in these experiments to start with. This only obscures the results, and makes it more difficult for the reader to take away important knowledge.

There is also no discussion of how the optimization is actually happening. In particular, what hyperparameters are being used (eg. optimizers, learning rates, etc.) for each of these methods? Notably, the accuracy of FedAvg on CIFAR-10 seems extremely low (especially in contrast to other works on FL and FedAvg, eg. [Reddi et al., 2021]).

On a more speculative note: It is not clear what is happening in the IID setting with respect to layer matching. One would naively think that in such settings, all the clients' layers would match according to their depth (ie. each client would learn roughly the same model), in which case there seems to be little to no benefit of this layer matching approach. Thus, it is unclear why centralized training does so poorly in such settings (and in fact, it is unclear what a single iteration refers to in the centralized setting).

### Minor comment

I would urge the authors to avoid speculative claims about theory. For example, the authors cite an entire paper and say that the convergence of their algorithm can be proven using similar techniques. This is a statement that really cannot be verified by a reader (save for perhaps the authors of the work being cited), and as such should generally be avoided.

###

---

### Official Review · Reviewer_XXWP · 2022-10-18
**interesting work**

This paper proposes a novel layer-matching approach that clients can collectively build a large and deep model by aggregating small client-side models without degrading model performance.

The paper is well-structured and easy to follow. It contains a detailed description of the proposed matching strategy. Experimental studies are also conducted to validate the proposed method.

Several comments:
1.  The stacked layer assumption may limit the application of the proposed method.
2.  It seems like the algorithm and the experimental settings do not consider partial participation in cross-device FL. As you mentioned, the proposed method is partially motivated by the communication and computation resources on the mobile device, which is usually concluded as the cross-device setting. Then how effective is your proposed algorithm when not all the clients can participate the training process in each round?
3.  	In lines 191-192, in what case can the FL achieve the convergence rate of $O(1/T)$? Does it need to introduce new assumptions for the convergence analysis?

---

### Decision · Program_Chairs · 2022-10-20

Accept (Poster)